# Evidence for vesicle-mediated antigen export by the human pathogen *Babesia microti*

Jose Thekkiniath[1], Nicole Kilian[1], Lauren Lawres[1], Meital A Gewirtz[1], Morven M Graham[3], Xinran Liu[2,3], Michel Ledizet[4], Choukri Ben Mamoun[1]

The apicomplexan parasite *Babesia microti* is the primary agent of human babesiosis, a malaria-like illness and potentially fatal tick-borne disease. Unlike its close relatives, the agents of human malaria, *B. microti* develops within human and mouse red blood cells in the absence of a parasitophorous vacuole, and its secreted antigens lack trafficking motifs found in malarial secreted antigens. Here, we show that after invasion of erythrocytes, *B. microti* undergoes a major morphogenic change during which it produces an interlacement of vesicles (IOV); the IOV system extends from the plasma membrane of the parasite into the cytoplasm of the host erythrocyte. We developed antibodies against two immunodominant antigens of the parasite and used them in cell fractionation studies and fluorescence and immunoelectron microscopy analyses to monitor the mode of secretion of *B. microti* antigens. These analyses demonstrate that the IOV system serves as a major export mechanism for important antigens of *B. microti* and represents a novel mechanism for delivery of parasite effectors into the host by this apicomplexan parasite.

# Introduction

Human babesiosis is an emerging tick-borne disease endemic in the United States and increasingly reported in other parts of the world, including Asia and Europe (Chen et al, 2015; Westblade et al, 2017; Ben Mamoun & Allred, 2018). Several species of *Babesia* have been shown to cause infection in humans with *Babesia microti* accounting for the large majority of clinical cases reported worldwide (Krause, 2019). In susceptible patients, *B. microti* infection can result in severe symptoms, including respiratory distress, splenic rupture, and renal failure (Leiby, 2011). The mortality rates associated with babesiosis infections range between 6 and 21%. Furthermore, severe infections and end-organ complications may develop in up to 57% of immunocompromised patients (Westblade et al, 2017).

*B. microti* is a member of the phylum Apicomplexa, which encompasses other important human pathogens, including *Plasmodium* parasites, the causative agents of malaria (Cornillot et al, 2012, 2016; Garg et al, 2014). The parasite develops and multiplies within red blood cells of its mammalian host to cause the pathological symptoms associated with babesiosis (Westblade et al, 2017; Ben Mamoun & Allred, 2018). However, unlike malaria parasites, which maintain a parasitophorous vacuole membrane throughout their intraerythrocytic development, *B. microti* has a transient parasitophorous vacuole that forms immediately after invasion but rapidly disintegrates as the parasite develops within the host cell (Rudzinska et al, 1976). Furthermore, structures such as the Maurer's clefts and the tubulovesicular network, previously shown to be present in the cytoplasm of *Plasmodium falciparum*–infected erythrocytes, are not found in *B. microti*–infected erythrocytes (Marti et al, 2005; Lanzer et al, 2006; Pelle et al, 2015; Cornillot et al, 2016; de Koning-Ward et al, 2016; Sherling & van Ooij, 2016). Proteomic, immunoproteomic, and genomic efforts used to search for major antigens of the parasite, which could be used as vaccines or diagnostic biomarkers, identified several antigens of *B. microti* that trigger a strong host immune response in mice and humans (Molestina et al, 2002; Cornillot et al, 2016; Silva et al, 2016; Elton et al, 2018). Among these, BmGPI12 (BmSA1) has been shown to trigger the strongest IgM and IgG responses and has been successfully used as a diagnostic marker of active *B. microti* infection (Thekkiniath et al, 2018). Although BmGPI12 and several other *B. microti* antigens carry an N-terminal signal peptide, sequence analysis showed that they lack canonical PEXEL (*Plasmodium* export element) or PEXEL-like motifs found in secreted proteins of other apicomplexan parasites such as *P. falciparum* and *Babesia bovis* (Boddey et al, 2016; Gallego-Lopez et al, 2018). This suggests that *B. microti* uses a novel trafficking mechanism for export of proteins into the host. Here, we show that vesiculation at the parasite plasma membrane (PPM) of *B. microti* produces an interlacement of connected vesicles in the host erythrocyte that is responsible for the export of at least two secreted antigens including BmGPI12 to the host erythrocyte and subsequently to mouse plasma.

---

[1]Department of Internal Medicine, Section of Infectious Diseases, Yale School of Medicine, New Haven, CT, USA   [2]Department of Cell Biology, Yale School of Medicine, New Haven, CT, USA   [3]Center for Cellular and Molecular Imaging Electron Microscopy Core Facility, Yale School of Medicine, New Haven, CT, USA   [4]L2 Diagnostics, Limited Liability Corporation, New Haven, CT, USA

Correspondence: choukri.benmamoun@yale.edu

# Materials and Methods

## Ethical clearance

All animal experimental protocols followed Yale University institutional guidelines for care and use of laboratory animals and were approved by the Institutional Animal Care and Use Committee at Yale University.

## Parasite strains

*B. microti* isolates LabS1 and PRA99 used in this study (Cornillot et al, 2016) were maintained in either *rag2*$^{-/-}$ knockout (B6.129S6-*Rag2*$^{tm1Fwa}$ N12) or SCID (severe combined immunodeficiency) CB17/Icr-*Prkdc*$^{scid}$/IcrIcoCrl mice as previously described (Lawres et al, 2016). In these mice, injection with $10^6$ infected red blood cells results in increasing parasitemia over time reaching a plateau after day 13 at ~50% parasitemia.

## Antibodies

Polyclonal antibodies used in this study were raised in rabbits after injection with purified recombinant BmGPI12 (amino acids 1–302); BmIPA48 (Genebank ID: XP_021338473; EupathDB ID: BMR1_03g00947) peptide CNKIKTDGGKVDSNS; and BmRON2 (Genebank ID: XP_021338832.1; EupathDB ID: BMR1_03g04695) peptide NKIKTDGGKVDSNS. Other antibodies against BmGPI12 used in this study (Figs S2 and S3) include three polyclonal antibodies against peptide 93–107 (AKSKLNKLEGESHK), 177–190 (KVSENLKDDEASAT), and 290–304 (AHRKAENLDVDDTL), and one monoclonal antibody, 5C11. Anti–TER-119 mouse monoclonal antibody (Invitrogen) was used to stain the plasma membrane of mouse erythrocytes.

## Plasma collection and fractionation studies

Blood from uninfected or *B. microti*–infected animals was collected by cardiac puncture and stored in tubes containing $K_2$EDTA (dipotassium ethylenediaminetetraacetate) solution. The samples were subjected to centrifugation at 1,300 rpm (200$g$) for 20 min at room temperature, plasma transferred to new tubes, the cell pellets washed twice in PBS containing 1% saponin, incubated on ice for 30 min, and spun at 9,300$g$ for 10 min at 4°C. The resulting supernatants (hemolysate) were collected, and the remaining pellets (uninfected) or parasite (infected) fractions were washed twice with PBS and spun at 9,300$g$ for 10 min at 4°C. Plasma (S), hemolysate (H), and pellet (P) fractions were mixed with Laemmli buffer, separated on SDS–PAGE, and analyzed by immunoblotting.

## Isolation of vesicles from plasma

Membrane vesicles were isolated from the plasma of uninfected mice or mice infected with *B. microti* by sequential centrifugations using established methods for isolation of exosomes with the following modifications (Baranyai et al, 2015). Briefly, 400 $\mu$l plasma from animals was diluted with 5 ml PBS and centrifuged at 500$g$ for 30 min and then at 16,000$g$ for 45 min to remove microvesicles. Vesicles were pelleted with ultracentrifugation (UC) at 120,000$g$ for 14 h at 4°C using a Sorvall MTX 150 Micro-Ultracentrifuge with an S52-ST Swinging-Bucket Rotor (Thermo Fisher Scientific). The resulting pellet (P1), was collected and the supernatant was spun again using the same procedure. The resulting pellet (P2) and supernatant (Us) fractions were collected.

## Immunoblot analysis

Equal volumes (10 $\mu$l) of plasma (before UC), supernatant (Us), and pellet (Up) fractions obtained after UC were analyzed by immunoblotting. The Us fractions (1 ml) were first concentrated using 20% trichloroacetic acid and resuspended in 200 $\mu$l of Laemmli buffer before electrophoresis. The pellet fractions (P1 and P2) were combined in 200 $\mu$l of Laemmli buffer (Up) before electrophoresis. Samples were loaded on 10% SDS–PAGE gel and transferred to nitrocellulose membranes. The membranes were blocked in 5% milk in blocking buffer and incubated with primary antibodies (anti-BmGPI12 [1:250], anti-BmIPA48 [1:100], anti-BmRON2 [1:100], anti–TER-119 [1:500], and preimmune sera [1:250 dilution]) overnight at 4°C, washed in TBS-T, and incubated with HRP–conjugated antirabbit IgG (1:10,000 dilution) for 1 h. After additional washes, the membranes were incubated with ECL Western blotting detection reagents (GE Healthcare) and exposed to a X-ray film using Kodak autoradiography. The same method was used for immunoblotting of parasite and erythrocyte membrane (P), erythrocyte cytoplasm or hemolysate (H), and plasma or supernatant (S) fractions.

## Immunofluorescence assay

Thin blood smears from uninfected or *B. microti*–infected mouse blood were prepared on cover slides (12-542-80 12 CIR.-1; Fisherbrand) and transferred to a 24-well plate (Costar, Corning Incorporated). Cells were fixed in 1% formaldehyde (Thermo Fisher Scientific) diluted in PBS at 37°C for 30 min, washed three times in PBS, and then incubated for 30 min in a blocking buffer (5% heat-inactivated fetal bovine serum [16000-044; Gibco], 5% normal goat serum [16210-072; Gibco-BRL], and 0.1% saponin [10% stock solution in PBS] [S7900-100G; Sigma-Aldrich]) at 37°C. To label the membrane of mouse erythrocytes, FITC-conjugated antimouse TER-119 (BioLegend) was used at a 1:1,000 dilution. To detect *B. microti* antigens, polyclonal and monoclonal antibodies against BmGPI12, BmIPA48, or BmRON2 were used as described above. Coverslips were incubated with the primary antibodies either overnight at 4°C or for 1 h at 37°C and washed three times with wash buffer. Secondary antibody (goat anti-rabbit IgG (H+L) rhodamine conjugate [Invitrogen]) was then added at 1:1,000 at 37°C for 1 h after which the cells were washed three times in wash buffer and then in PBS. Coverslips were then mounted on sandblasted single-frosted precleaned microscope slides (Thermo Fisher Scientific) using Prolong Gold antifade reagent supplemented with DAPI (P36935; Invitrogen by Thermo Fisher Scientific) and incubated at room temperature in the dark overnight, before they were examined with the Nikon ECLIPSE TE2000-E. A 100× oil immersion objective was

used for image acquisition. Excitation at 465–495 nm (bandpass 515–555 nm, dichroic mirror 505 nm) was used to detect FITC-positive cells; 510–560 nm (longpass 610 nm, dichroic mirror 575 nm) was used to detect rhodamine-positive cells; and 340–380 nm (bandpass 435–485 nm, dichroic mirror 400 nm) was used for DAPI. The images were acquired using MetaVue with 1,392 × 1,040 pixel as chosen image size.

### Sample preparation for electron microscopy cryosectioning

Samples of *B. microti*–infected erythrocytes were fixed in 4% PFA in PBS for 30 min at room temperature followed by further fixation in 4% PFA at 4°C for 1 h, rinsed in PBS, and resuspended in 10% gelatin. Chilled blocks were trimmed and placed in 2.3 M sucrose overnight on a rotor at 4°C, transferred to aluminum pins, and frozen rapidly in liquid nitrogen. The frozen blocks were cut on a Leica Cryo-EMUC6 UltraCut, and 65-nm-thick sections were collected using the Tokoyasu method (Tokuyasu, 1973) and placed on carbon/formvar–coated grids and floated in a dish of PBS for immunolabeling.

### High-pressure freezing and freeze-substitution Epon section and labeling

Samples fixed in 4% PFA were frozen using a Leica HMP100 at 2,000 psi. The frozen samples were then freeze-substituted using a Leica Freeze AFS unit starting at –95°C using 1% osmium tetroxide, 1% glutaraldehyde, and 1% water in acetone for 10 h, warmed to –20°C for 12 h and then to 4°C for 2 h. The samples were rinsed in 100% acetone and infiltrated with Durcupan resin (Electron Microscopy Science) and baked at 60°C for 24 h. Hardened blocks were cut using a Leica UltraCut UC7, and 60-nm sections were collected on formvar/carbon–coated nickel grids.

### Immunolabeling of resin sections

Grids were placed section side down on drops of 1% hydrogen peroxide for 5 min, rinsed, and blocked for nonspecific binding on 3% bovine serum albumin in PBS containing 1% Triton-X for 30 min. They were then incubated with a primary antibody rabbit anti-BmGPI12 or anti-BmIPA48 1:100 overnight, rinsed in buffer, and then incubated with the secondary antibody 10 nm protein A gold (UtrechtUMC) for 30 min. The grids were rinsed and fixed using 1% glutaraldehyde for 5 min, rinsed well in distilled water, and contrast-stained using 2% uranyl acetate and lead citrate. The grids were all viewed in an FEI Tencai Biotwin TEM at 80 kV. Images were taken using Morada CCD and iTEM (Olympus) software.

### Image processing

FIJI (imagej.net/Fiji) and Microsoft PowerPoint (Microscoft Corporation) was used to analyze and prepare raw images of Giemsa smears, EM images, and fluorescence images for presentation.

Diameter and length of EM structures were determined via the line tool in FIJI.

## Results

### BmGPI12, the major secreted antigen of *B. microti*, is associated with parasite-derived membrane vesicles

The immunodominant BmGPI12 of *B. microti* is encoded by a member of the *bmn* multigene family and is one of the most highly expressed genes of the parasite during its development within red blood cells (Lodes et al, 2000; Cornillot et al, 2012, 2016; Silva et al, 2016). Consistent with the secretion of BmGPI12 from the parasite into the red blood cytoplasm and subsequently into the host (Luo et al, 2011, 2012; Cornillot et al, 2016; Thekkiniath et al, 2018), immunoblot analyses using anti-BmGPI12 antibodies on blood collected from mice and fractionated to collect plasma (S), erythrocyte cytoplasm (H), and membrane (P) fractions showed the presence of BmGPI12 in all three fractions from animals infected with *B. microti* strains (LabS1 or PRA99) (Fig 1A), but not from uninfected animals (Fig 1A). As a control, immunoblot analyses conducted using a monoclonal antibody against the mouse erythrocyte membrane protein, TER-119 (glycophorin A–associated protein [Ly-76]), identified this protein in the membrane (P) fractions of both uninfected and *B. microti*–infected erythrocytes (Kina et al, 2000), but not in the plasma or erythrocyte cytoplasm fractions (Fig 1A). As expected, antibodies against the *B. microti* rhoptry neck protein BmRON2, a highly conserved protein among apicomplexan parasites that localizes to the parasite apical end (Ord et al, 2016) and released upon egress of daughter parasites, identified the protein in the membrane and plasma fractions, but not in the erythrocyte cytoplasm (Fig S1). This finding is consistent with BmRON2 association with the parasite during its intraerythrocytic development and its release after the rupture of the infected erythrocyte. As a control, preimmune sera were used to analyze the fractions from uninfected and *B. microti*–infected erythrocytes, and no signal could be detected (Fig S1).

Fluorescence microscopy analyses identified BmGPI12 in both the cytoplasm and plasma membrane of the parasite as well as in the cytoplasm of the infected erythrocyte in well-defined dendrite-like structures and distinct foci (Figs 1B and S2). These structures are reminiscent of membranous extensions often seen in Giemsa-stained blood smears of *B. microti*–infected erythrocytes at different stages of parasite development (representative images of LabS1) (Fig 2A). Interestingly, these membrane extensions are also observed in *Babesia duncani*–infected human red blood cells (Fig S3) maintained in cell culture as previously described (Abraham et al, 2018). Blood smears prepared from four *B. microti*–infected mice showed that the parasite undergoes major morphogenic changes throughout its intraerythrocytic life cycle that includes ring-shaped forms, rings with dendrite-like tubulovesicular structures (tubes of vesicles [TOVs]) and to a lesser extent dividing parasites (tetrads) and tetrads with TOVs (Fig 2B and C). The proportion of parasites with TOV structures remained the same with increasing parasitemia

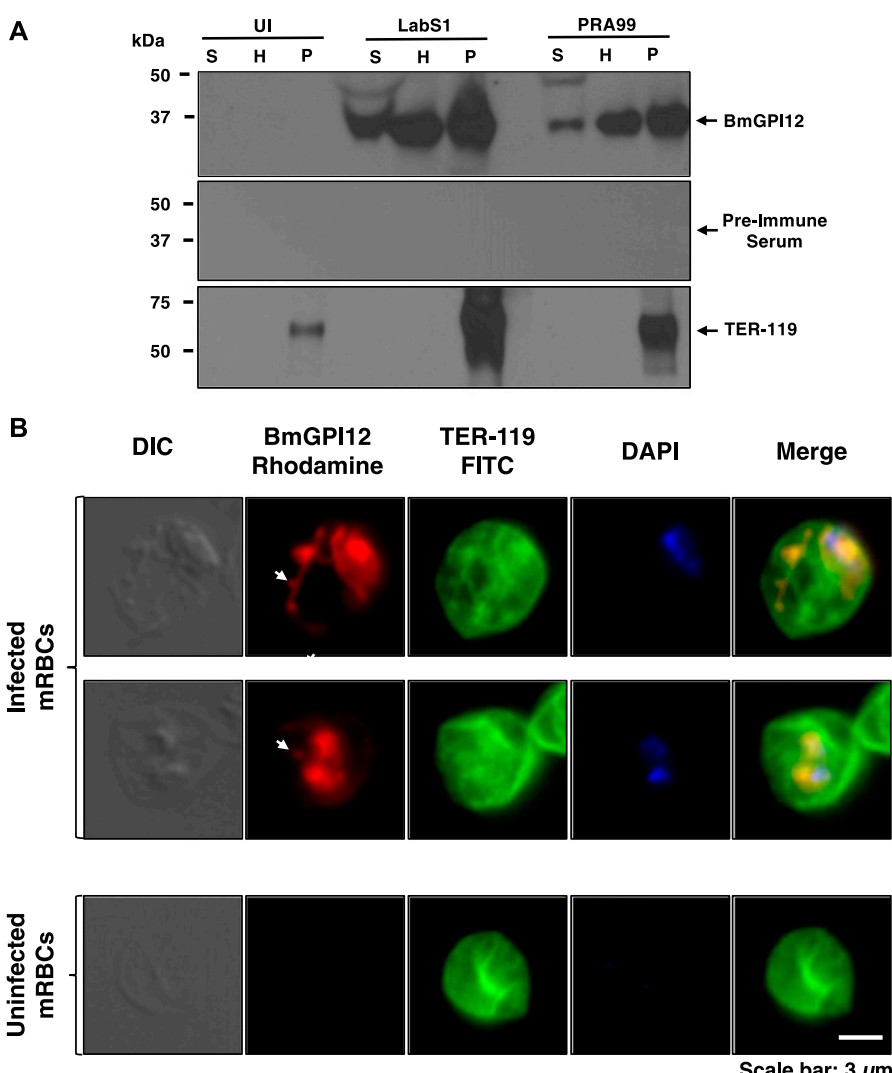

**Figure 1. BmGPI12 is secreted into the erythrocyte cytoplasm and subsequently into the extracellular environment of the *B. microti*–infected erythrocyte.**
**(A)** Immunoblotting analysis using preimmune (PI) and anti-BmGPI12 immune rabbit sera on fractions of uninfected erythrocytes (UI) or erythrocytes infected with either PRA99 or LabS1 strain of *B. microti*. In uninfected erythrocytes, the P fraction consists primarily of erythrocyte membranes. In *B. microti*–infected erythrocytes, the P fraction includes both erythrocyte membranes and protein extracts from isolated parasites. The erythrocyte membrane protein TER-119 (52 kD) was detected only in the P fractions from uninfected and *B. microti*–infected red blood cells using anti–TER-119 monoclonal antibody. **(B)** Immunofluorescence assay on mouse erythrocytes infected with the PRA99 strain of *B. microti*. BmGPI12 was labeled with anti-BmGPI12 polyclonal antibodies and could be observed within the parasite cytoplasm, the PPM as well as in the erythrocyte cytoplasm and within IV and TOVs (indicated by arrowheads). Anti–TER-119 monoclonal antibody was used to label the plasma membrane of the infected mouse erythrocytes. The DAPI staining was applied to verify the presence of parasites within the erythrocytes by labeling parasitic nuclear DNA. Staining of control uninfected red blood cells using the same antibodies is shown (lower panel). Scale bar: 3 *μ*m. H, hemolysate; mRBC, mouse red blood cells; P, membrane fractions; S, mouse plasma.

in infected animals, suggesting that they represent distinct development stages of *B. microti* (Fig 2C).

### Vesicle-mediated export of BmGPI12 antigen into the cytoplasm of *B. microti* erythrocytes and host plasma

To investigate the nature of the TOVs at the ultrastructural level, electron microscopy analyses of ultrathin sections of erythrocytes from *B. microti*–infected mice were conducted. These analyses revealed tubes of connected vesicles as well as individual vesicles (IVs) in the cytoplasm of infected cells (Figs 3A and B, and S4), but not in the uninfected cells (Fig 3C). The diameter of IVs is ~0.110 *μ*m ± 0.0052 *μ*m (Mean ± SEM), whereas the length of TOVs ranges between ~0.405 *μ*m ± 0.056 *μ*m (Mean ± SEM) and 0.900 *μ*m, depending on the sections. Interestingly, detailed analysis of these structures revealed that the TOVs emerge directly from the PPM and extend into the infected erythrocyte (Figs 3B and S4B). Both the cytoplasm of the parasite and the content of the vesicles and tubules share the same electron density (Figs 3A and B, and

S4), further demonstrating that the structures are of parasite origin.

To determine whether the IV and TOV structures identified by electron microscopy represent the same BmGPI12-positive structures detected by fluorescence microscopy, immunoelectron microscopy (IEM) analyses were performed on *B. microti*–infected murine erythrocytes using anti-BmGPI12 antibodies coupled to 10-nm gold particles. IEM analyses showed that BmGPI12 localizes to the PPM as well as to the IVs and the TOVs (Fig 4A and B).

### Export of BmGPI12-containing vesicles from *B. microti*–infected erythrocytes

The finding that *B. microti* produces IVs and TOVs inside infected erythrocytes and that BmGPI12 is associated with these structures led us to further investigate whether this protein is secreted into host plasma via a vesicle-mediated secretory mechanism or as a free antigen. This was achieved by subjecting plasma samples from *B. microti*–infected mice to UC at 120,000*g* to separate fractions

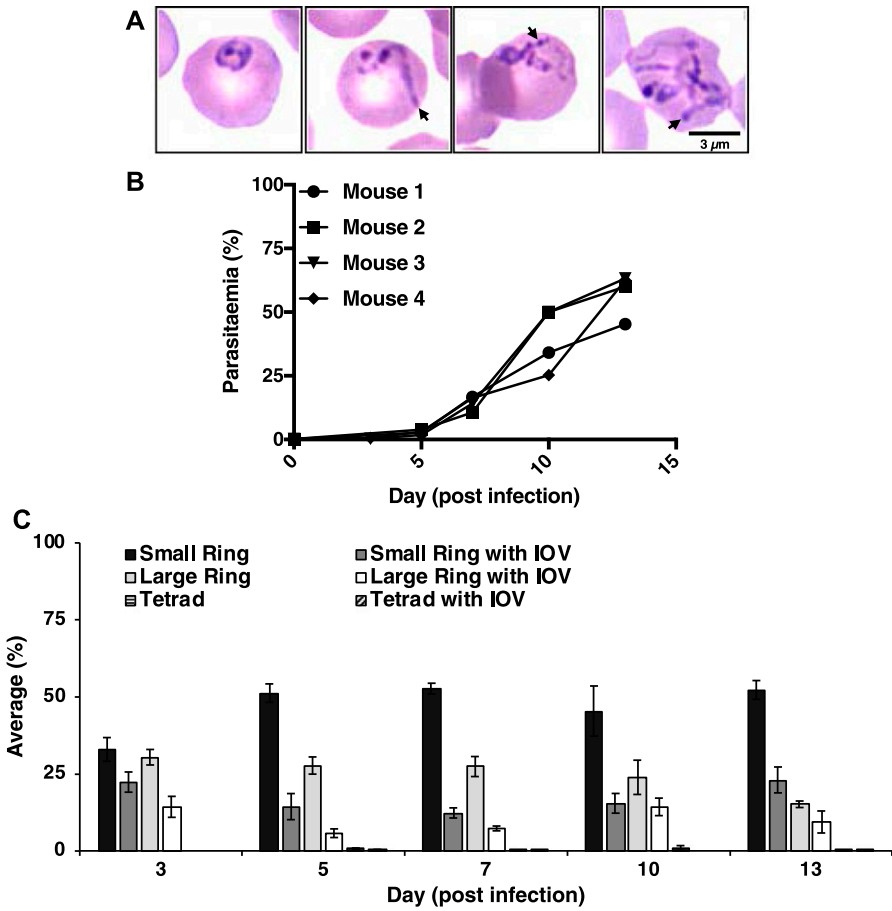

**Figure 2. *B. microti* develops an IOV system in the cytoplasm of the infected erythrocytes.**
**(A)** Representative images of Giemsa-stained blood smears from *B. microti* (LabS1)–infected erythrocytes. Long membranous structures within the erythrocyte cytoplasm are indicated by arrowheads. Scale bar: 3 µm. **(B, C)** Analysis of blood from four *B. microti*–infected SCID mice over a 13-d period after infection with the LabS1 strain. **(B)** Parasitemia levels in individual mice. For each sample, a total of 5,000 erythrocytes were analyzed. **(C)** Proportion of each morphological form detected in the blood smears at days 3, 5, 7, 10, and 13 postinfection. A total of 20 images were analyzed per smear at a given day (Mean ± SEM).

containing membrane-associated structures, including vesicles and tubules (Up) from those containing free proteins (Us) (Fig 4C). As shown in Fig 4D, BmGPI12 was found in both the Up and Us fractions, suggesting that it is released into host plasma as a membrane-associated protein, but could also be found as a free protein, most likely because of cleavage of the GPI-anchor by plasma enzymes. As expected, the host protein TER-119, which associates exclusively with the erythrocyte membrane, was not found in the Up or Us fractions (Fig 4D). Analysis of the membrane fractions (Up) by IEM identified BmGPI12 associated with vesicles and tubular structures with similar sizes as those observed inside the infected erythrocytes (Fig 4E and F).

### Evidence for vesicular-mediated export of the immunodominant antigen BmIPA48 in *B. microti*

To determine whether the vesicular system used by *B. microti* for secretion of BmGPI12 is also used by the parasite to export other antigens, we examined the cellular distribution of another *B. microti* antigen BmIPA48, which was previously shown to trigger strong IgM and IgG response in infected outbred mice (Silva et al, 2016). BmIPA48 encodes a 48-kD antigen with an N-terminal signal peptide but no GPI-anchor motif or transmembrane domains (Silva et al, 2016). As shown in Fig 5, BmIPA48 is expressed in the parasite, secreted into

the erythrocyte cytoplasm, and then released into the host plasma (Fig 5A). Similar to BmGPI12, fluorescence microscopy shows association of the antigen with discrete foci in the infected red blood cell (Fig 5C). However, unlike BmGPI12, analysis of the distribution of BmIPA48 following UC showed the presence of the protein exclusively in the vesicle-containing fraction (Up fraction) (Fig 5B). Consistent with these findings, IEM analyses of *B. microti*–infected erythrocytes showed the presence of BmIPA48 inside vesicles found both in the parasite cytoplasm as well as secreted by the parasite into the erythrocyte cytoplasm (Fig 5D and E).

## Discussion

In this study, we report the first evidence that the human pathogen *B. microti* uses a novel mechanism to export its proteins into the host erythrocyte and subsequently into host plasma. This system uses a network of parasite-derived dendrite-like membrane branches consisting of connected vesicles. We show that at least two immunodominant antigens of *B. microti*, BmGPI12 and BmIPA48, are exported by the parasite via this mechanism.

A recent study estimated that ~398 proteins may be secreted by *B. microti* during its development within mammalian erythrocytes (Silva et al, 2016). Some of these proteins may be exported into the

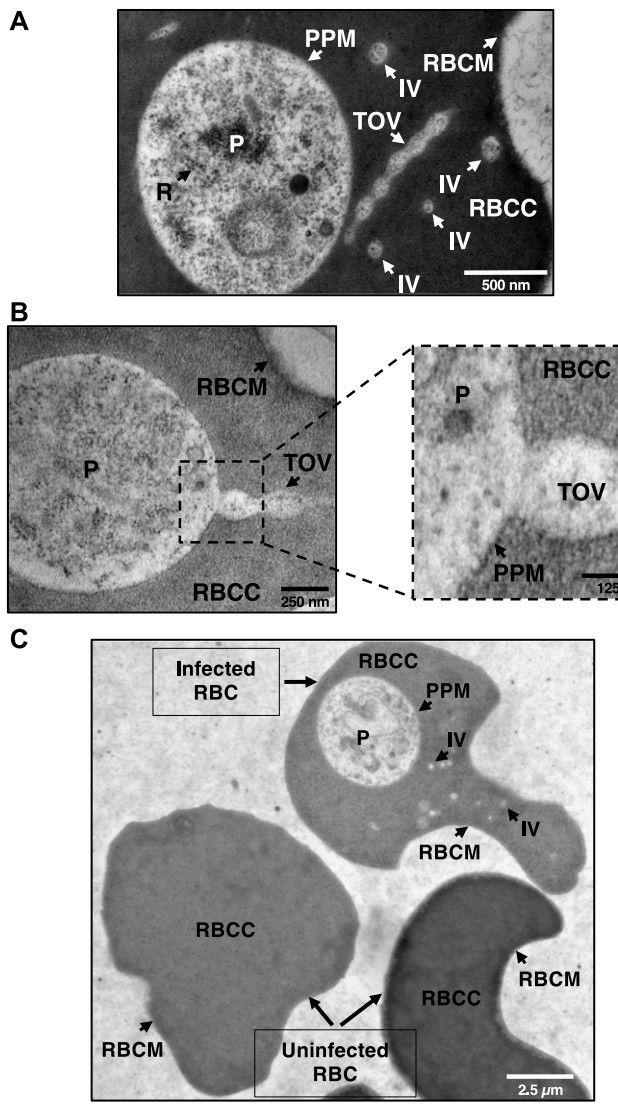

**Figure 3. *B. microti* develops an IOV system in the cytoplasm of the infected erythrocytes.**
**(A)** EPON-embedded LabS1-infected erythrocytes revealed the presence of the IOV system in the erythrocyte cytoplasm. The IOV system contains the same electron-dense structures as the parasite, indicating that these structures are of parasitic origin. **(A, B)** Various structures of parasites and erythrocytes are shown by arrows (A and B). **(C)** Comparison of ultrathin sections of EPON-embedded infected and uninfected erythrocytes (C). Scale bars: 500 nm (A), 250 nm and 125 nm (B), 2.5 μm (C). P, parasite; R, ribosomes; RBCC, red blood cell cytoplasm; RBCM, red blood cell membrane.

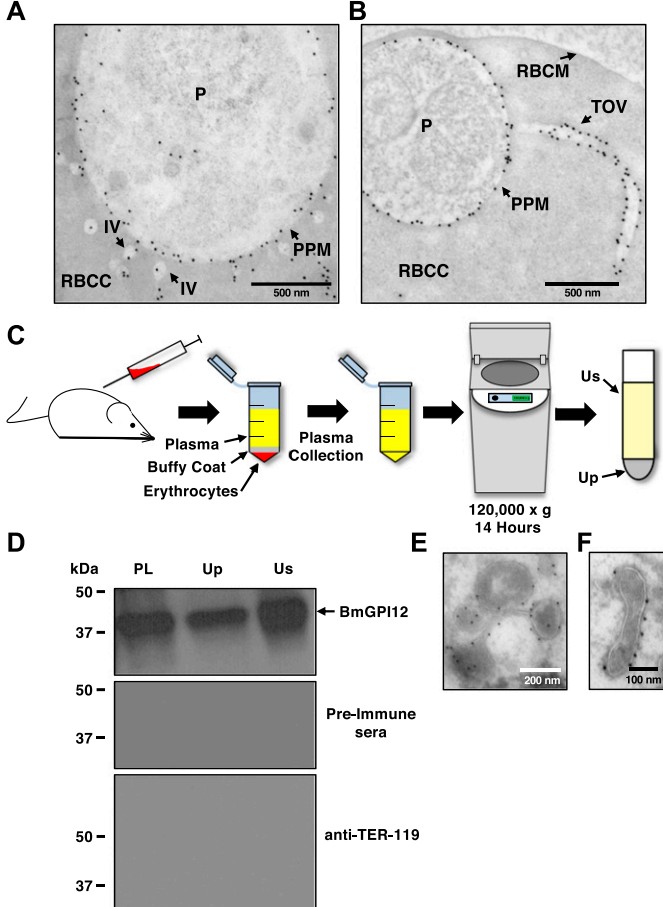

**Figure 4. BmGPI12 is localized to the PPM and associated with vesicles and tubules.**
**(A, B)** Immunoelectron microscopic analysis of *B. microti* LabS1–infected mouse erythrocytes. Ultrathin sections of high-pressure frozen and Durcupan resin–embedded infected erythrocytes were immunolabeled with anti-BmGPI12 polyclonal antibodies (amino acid 1–302). Scale bars: 500 nm (A, B). **(C)** Schematic diagram showing the steps in the UC of plasma samples collected from *B. microti*–infected mice. **(D)** Immunoblot analyses using preimmune (PI) serum, and anti-BmGPI12 or anti–TER-119 antibodies on either intact plasma (PL) collected from mice infected with *B. microti* LabS1 strain or on two fractions (supernatant: Us and pellet: Up) of plasma after UC at 120,000*g*. **(E, F)** Immunoelectron microscopic analysis of the plasma membrane fraction (Up) from mice infected with *B. microti* LabS1 using anti-BmGPI12 antibodies coupled to 10-nm gold particles. Scale bars: 200 nm (E), 100 nm (F). P, parasite; RBCC, red blood cell cytoplasm; RBCM, red blood cell membrane.

erythrocyte cytoplasm or erythrocyte membrane where they may function to modulate the host cell cytoskeleton or to facilitate uptake of nutrients. Others might be further exported into the host plasma to modulate the host response or effect other changes beneficial to the parasite. Both BmGPI12 and BmIPA48 contain an N-terminal signal peptide but lack specific motifs such as the PEXEL motif found in other apicomplexan parasites and associated with secretion of proteins into the host (Marti et al, 2005; Lanzer et al, 2006; Pelle et al, 2015; Cornillot et al, 2016; de Koning-Ward et al, 2016; Sherling & van Ooij, 2016). In fact, all predicted secreted proteins of *B. microti* lack such a motif (Silva et al, 2016), suggesting

that this parasite might have evolved a novel mode of protein export. Interestingly, unlike *P. falciparum* where the PEXEL motif plays an important role in the trafficking of parasite proteins across the parasitophorous vacuole that separates the parasite from the erythrocyte cytoplasm, *B. microti* spends most of its intra-erythrocytic development without a vacuole, thus eliminating the need for a translocon to export proteins into the host cytoplasm. Consistent with this model, analysis of the *B. microti* genome shows the lack of orthologs of most components of the malaria translocon (Cornillot et al, 2012; Silva et al, 2016).

Our electron microscopy analyses of ultrathin sections of *B. microti*–infected erythrocytes showed the presence of interlacement of vesicles (IOV) consisting of IVs and TOVs with a

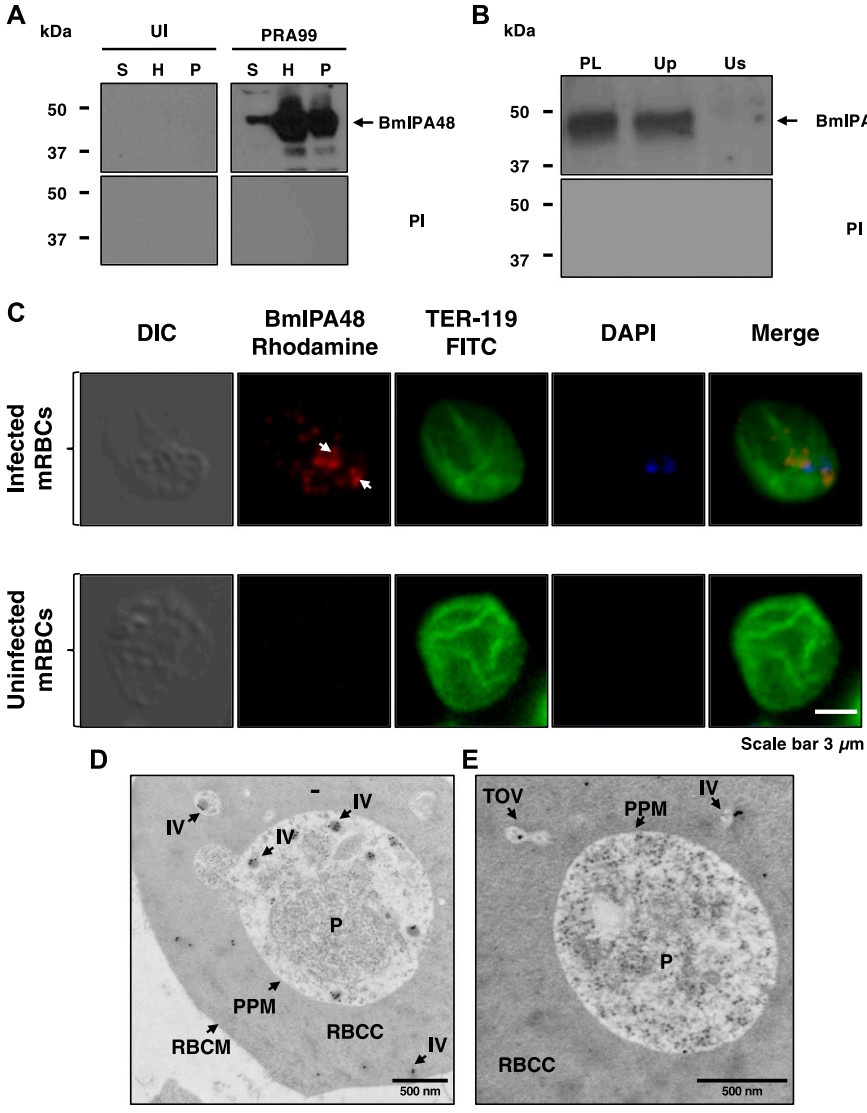

**Figure 5.  Vesicle-mediated secretion of BmIPA48 antigen by *B. microti*.**
**(A)** Distribution of BmIPA48 in the plasma (S), hemolysate (H), and membrane (P) fractions isolated from blood of uninfected or *B. microti* (PRA99)–infected erythrocytes was determined by Western blotting using polyclonal antibodies against BmIPA48 (48 kD). Preimmune (PI) sera were used as control. **(B)** Immunoblot analysis using preimmune and anti-BmIPA48 antibodies on either intact plasma (PL) collected from mice infected with *B. microti* PRA99 strain or on two fractions (supernatant: Us and pellet: Up) of plasma after UC at 120,000*g*. **(C)** Immunofluorescence assay using anti-BmIPA48 in PRA99-infected erythrocytes. BmIPA48 was labeled with polyclonal antibodies and could be detected within the parasite and in discrete IVs within the cytoplasm of the infected erythrocyte. Anti–TER-119 monoclonal antibody was used to label the plasma membrane of mouse erythrocytes and DAPI was used to stain the parasite nucleus. Staining of control uninfected red blood cells is shown. Scale bar: 3 *µ*m. **(D, E)** Representative images of immunoelectron micrographs of *B. microti* LabS1–infected mouse erythrocytes. Ultrathin sections of high-pressure frozen and Durcupan resin–embedded infected erythrocytes were immunolabeled with anti-BmIPA48 antibodies coupled to 10-nm gold particles. Scale bars: 500 nm (D and E). mRBC, mouse red blood cells; P, parasite; RBCC, red blood cell cytoplasm; RBCM, red blood cell membrane.

diameter of ~0.110 *µ*m ± 0.0052 *µ*m. Giemsa staining showed that the TOVs can vary in length between infected erythrocytes with some extensions measuring several micrometers in diameter. Further analysis of the blood smears showed the presence of TOVs throughout the life cycle of the parasite and suggests that production of filamentous forms represent a distinct morphogenic event in the development of the parasite. Interestingly, analysis of cryosections showed that the IOV system of *B. microti* is of a composition similar to that of the cytoplasm of the parasite. An enlargement of a section near the PPM in Fig 3B showed a tubule consisting of two vesicles directly emerging out of the parasite membrane. This distinguishes the system from the Tubo-Vesicular Membrane system previously described in *P. falciparum*, which has been shown to emerge from the parasitophorous vacuolar membrane of the malaria parasite. Future analyses of *B. microti*–infected erythrocytes using three-dimensional electron microscopy or focus ion beam scanning electron microscopy could shed more light on the connection between vesicles and plasma membranes of the parasite and possibly the host erythrocyte.

In this study, we have also employed fractionation methods used to characterize the secreted vesicles produced by *B. microti*. Using this approach, we found that BmGP12 could readily be detected in both the soluble and membrane fractions after centrifugation of plasma samples, whereas BmIPA48 was found primarily in the vesicle-rich membrane fractions. The difference in the distribution of these two proteins could be due to their respective localization in the vesicles and tubules. Our IEM analyses showed that BmGPI12 is primarily found on the PPM and on the membranes of vesicles and tubules. Interestingly, the 10-nm gold particles could be found both inside and outside these vesicles and tubules. It is, therefore, likely that after vesicular release, BmGPI12 exposed outside the vesicles is cleaved and thus found in the soluble fraction, whereas the proteins still residing inside the vesicles remain associated with the membrane fraction (Fig 6). On the other hand, IEM analysis of BmIP48 showed the presence of the protein primarily inside the vesicles, consistent with their exclusive association with the membrane fraction. Interestingly, none of the vesicles could be detected on the outer layer of the erythrocyte membrane of

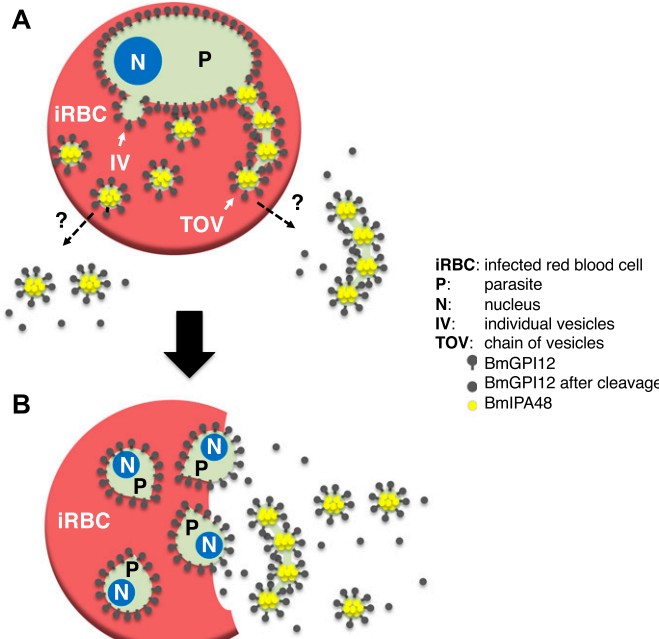

**iRBC**: infected red blood cell
**P**: parasite
**N**: nucleus
**IV**: individual vesicles
**TOV**: chain of vesicles
 BmGPI12
 BmGPI12 after cleavage
 BmIPA48

**Figure 6.** **Model of vesicular-mediated export of antigens by *B. microti* during development and after rupture.**
**(A)** Dashed lines indicate yet-to-be-identified vesicle export pathways during *B. microti* intraerythrocytic development (rings) before cell division. **(B)** Model of export of TOVs and IVs after parasite division and rupture of the infected erythrocyte. The tetrad (The Maltese cross) stage is a transient developmental stage rarely detected in infected animals.

infected cells, suggesting that the vesicles are either actively exported through an unidentified export machinery during parasite development within the host erythrocyte or released outside the red blood cell after parasite cell division (to form four daughter parasites) and subsequent rupture of the infected erythrocyte (see model in Fig 6). In *B. microti* tetrad forms (Maltese cross) are rarely detected in freshly isolated infected blood (Fig 2C), suggesting that the two cell division events that lead to formation of four daughter parasites from a single mother cell occur rapidly during the intraerythrocytic cycle and are followed by rapid exit of daughter parasites and rupture of the infected cell.

Noteworthy, filamentous structures can also be seen in *B. duncani*–infected human red blood cells both during development as well as in the tetrad form as filaments connecting daughter parasites (Fig S3). Although some previously published images of Giemsa-stained blood smears of other *Babesia* species (*Babesia divergens* or *B. bovis*) (Hildebrandt et al, 2013; Sevilla et al, 2018) or *Theileria annulata* (Kuhni-Boghenbor et al, 2012) show unusual protrusions extending from individual or dividing parasites, these structures are not similar to the filamentous forms seen in *B. microti*– and *B. duncani*–infected erythrocytes. It remains unknown whether this machinery is common to other piroplasms and what is its primary function in the physiology of these parasites.

In summary, the present study provides the first evidence that *B. microti* uses a novel mechanism of protein export to deliver proteins into the host erythrocyte and plasma. Understanding the molecular mechanism underlying this export process and the

components involved in such machinery might provide a unique opportunity to target the parasite and develop more effective therapies.

## Data Availability

Data and materials supporting the findings of this study are available from the corresponding author upon reasonable request.

## Supplementary Information

## Acknowledgements

We thank the staff at the Yale University CCMI Electron Microscopy Core Facility for their technical support. We thank A Abraham for her assistance with parasitemia determination. This work was funded by NIH grants AI123321 and R43AI136118 and in part supported by the NIH grant S10 OD020142.

### Author Contributions

J Thekkiniath: conceptualization, data curation, formal analysis, validation, investigation, visualization, methodology, and writing—original draft, review, and editing.
N Kilian: conceptualization, resources, formal analysis, validation, investigation, visualization, methodology, and writing—original draft, review, and editing.
L Lawres: investigation and writing—original draft.
M Gewirtz: validation, investigation, and writing—original draft, review, and editing.
MM Graham: investigation, methodology, and writing—original draft.
X Liu: investigation, methodology, and writing—original draft.
M Ledizet: resources.
C Ben Mamoun: conceptualization, resources, data curation, formal analysis, supervision, funding acquisition, validation, investigation, visualization, methodology, project administration, and writing—original draft, review, and editing.

### Conflict of Interest Statement

The authors declare that they have no conflict of interest.

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
