## [Reviewer comments · Life Science Alliance]

Life Science Alliance

Evidence for vesicle-mediated antigen export by the human pathogen *Babesia microti*

Jose Thekkiniath, Nicole Kilian, Lauren Lawres, Meital Gewirtz, Morven Graham, Xinran Liu, Michel Ledizet, and Choukri Ben Mamoun

DOI: <https://doi.org/10.26508/lsa.201900382>

Corresponding author(s): Choukri Ben Mamoun, Yale School of Medicine

Review Timeline:

Submission Date:	2019-03-14
Editorial Decision:	2019-04-23
Revision Received:	2019-05-23
Editorial Decision:	2019-06-04
Revision Received:	2019-06-05
Accepted:	2019-06-06

Scientific Editor: Andrea Leibfried

Transaction Report:

April 23, 2019

Re: Life Science Alliance manuscript #LSA-2019-00382-T

Dr. Choukri Ben Mamoun
Yale School of Medicine
Section of Infectious Diseases
15 York Street
Winchester Building WWW403D
New Haven, CT 6520

Dear Dr. Ben Mamoun,

Thank you for submitting your manuscript entitled "Evidence for vesicle-mediated antigen export by the human pathogen *Babesia microti*" to Life Science Alliance. The manuscript was assessed by expert reviewers, whose comments are appended to this letter.

As you will see, the referees acknowledge that the findings are potentially interesting. However, they also point out several concerns and have a number of suggestions on how to strengthen the manuscript, and I think that all of them should be addressed. In particular, the EM data from *B. microti* should be compared with at least two other strains of *Babesia* parasites as suggested by Reviewer #1. The further reaching question from Reviewer #2 on the selectivity of ToVs for certain cargo (point 1) however, would certainly strengthen the manuscript but it will not be mandatory to address this point experimentally.

Given these constructive comments, we would thus like to invite you to revise your manuscript for Life Science Alliance.

Thank you for this interesting contribution to Life Science Alliance. We are looking forward to receiving your revised manuscript.

Sincerely,

B. MANUSCRIPT ORGANIZATION AND FORMATTING:

Reviewer #1 (Comments to the Authors (Required)):

In their manuscript, the authors describe the ultrastructure of *Babesia microti* in the red blood cell. They observe a membranous network budding off from the parasite. While the paper seems straightforward there are quite a number of inconsistencies that need to be addressed prior to publication:

The authors need to compare EM level data from *B. microti* with data from at least two other *Babesia* parasites. Even by Giemsa staining it is very obvious that *B. microti* has a different intracellular appearance as most other *Babesia* species. This needs to be clearly and upfront communicated. The reported observation could also be just stretching events as seen in *P. falciparum* ring stages. This should be discussed.

The BmGPI12 staining with confocal needs another control antibody to convince this reviewer.

Other points:

Figure 1, Figure S1A and Figure 5A look like different exposures of the same blot. The authors should supply a similar exposed version of the different blots to diffuse possible concerns.

Figure 3B and Fig S3B are duplicates.

Figure 6B needs legend, how does export work?

Figure S1 misses 'S' in second panel

Authors need to mention that 3D EM would be beneficial to look at whole cell, e.g. FIB SEM or serial sections to investigate whether the vesicles are truly disconnected from the parasite body.

Page 4, there are more than 3 species causing disease in humans.

Page 12: the protein is NOT found in the plasma fraction according to Figure S2. Please clarify. The labelling of the blots is not ideal

Page 12: the Ord paper does not mention what it is referred at.

Page 14: what do the authors mean by 'erythrocyte environment'?

Page 17: short term culture is not described in the methods

Page 18: maybe a concluding sentence or two at end of discussion?

Reviewer #2 (Comments to the Authors (Required)):

The manuscript is interesting to explore how the antigen of *Babesia microti* is being transported to RBC membrane. However, I have some concerns in the manuscript as follows.

1. Two strains of *B. microti* were used in the present study. Furthermore, several knockout mice were used for maintaining of strains. However, I could find the reasons why two strains and knockout mice were necessary. Were there some differences in the development of parasitemia between immunocompetent and knockout mice, and results between two strains, respectively? The authors should add the information on these questions.
2. For the determination of parasitemia, the sentence must be mixed up with some other method (p.6, lines 10-14).
3. There was no data for "TER-119 was not identified in the plasma (p. 12, line 15) in Fig, 1B".
4. The authors described that BmIPA48 was found inside of vesicles (p.15, line 11). However, some gold particles were found mainly outside of IV indicated by arrow in the upper left of Fig.5E.
5. Maltese cross (Tetrad form) is a morphologic characteristic of *B. microti* and it appears in the early

stage (less than 50% of whole parasitized RBC) before peak parasites and disappears in the later stage of developmental course of parasitemia in BALB-c mice (Infection and Immunity, 71:411-417, 2003). Although tetrad forms and tetrad forms with IOV were seldom found in the present study (Fig. 2C), the model of antigen export was suggested for tetrad form only in Fig. 6 without any explanations. Is this model applicable to paired form (two parasites)? If so, what were data for this model? The authors should suggest a model based on the evidence.

Reviewer #3 (Comments to the Authors (Required)):

Thekkianiath et al present remarkable data suggesting novel mechanism of protein transport used by the intracellular parasite, *Babesia microti*. This parasite resides within the RBC but is not contained inside a vacuole. Its genome lacks homologs of the translocon machinery used for protein export by *Plasmodium falciparum*, a related Apicomplexan that also parasitizes RBCs but one that resides within a vacuole. The authors demonstrate, primarily through EM, that intracellular *B. microti* stages form a network of long tubes of incipient vesicles, termed ToV, that extrude from the parasite plasma membrane. Infected RBCs also contain vesicles containing parasite proteins. At least, two secreted proteins of *B. microti*, BP112 and IPA48, that elicit strong host antibody responses, are present in these ToVs and vesicles. Their immunolocalization is supported by cell-fractionation experiments. Localization of BP112 and IPA48 to ToV suggests that these structures are part of the parasite-encoded machinery for trafficking proteins to the host cell cytoplasm for eventual secretion. The EM data is compelling and supportive of the paper's conclusions. One minor weakness that can be addressed in the Discussion section, is the relevance of the observation that IPA48 is not detected in the vesicle-free fraction. In this reviewer's opinion, this could result from lower sensitivity of the anti-IPA48 antibody compared to the anti-GP112 antibody or the higher overall abundance of GP112. Additional comments below are meant as suggestions to the authors.

1) It would be useful to test if ToV and vesicles were selective in their cargo. For example, testing the presence in ToVs or vesicle of parasite cytoplasmic proteins. If ToVs and vesicles serve in polar transport of secreted proteins, then parasite cytoplasmic proteins would be expected to be excluded from these structures. Furthermore, such "selection of proteins" could imply that *B. microti* uses a novel signal to mark proteins destined for secretion. On the other hand, if ToVs and vesicles bud off from the parasite plasma membrane in a more indiscriminate fashion, then they may contain both proteins destined for secretion and ones meant to be cytoplasmic.

2) Is there any evidence for ToV merging with the RBC PM? It would be intriguing if the ToV directly deliver proteins to the RBC surface. What is the relationship between Bgp112-containing ToVs in Fig 4B and 4 E-F. Are the vesicles and tubes in Fig 4E-F thought to arise from the ToVs seen in Fig 4B?

3) Is ToV formation associated with a specific parasite stage? While this seems to be the motivation behind Fig 2C, the conclusion is not clearly stated.

4) Can the authors comment on how whether other *Babesia* species form TOVs?

Reviewer #1 (Comments to the Authors (Required)):

In their manuscript, the authors describe the ultrastructure of *Babesia microti* in the red blood cell. They observe a membranous network budding off from the parasite. While the paper seems straightforward there are quite a number of inconsistencies that need to be addressed prior to publication:

Major comments

The authors need to compare EM level data from *B. microti* with data from at least two other *Babesia* parasites. Even by Giemsa staining it is very obvious that *B. microti* has a different intracellular appearance as most other *Babesia* species. This needs to be clearly and upfront communicated. The reported observation could also be just stretching events as seen in *P. falciparum* ring stages. This should be discussed.

Response:

We thank the reviewer for the suggestions. We provide Giemsa- stained smears from *B. duncani*, which has been recently successfully cultured in human RBCs (Abraham et al., JBC 2018). The filamentous forms that we see in *B. microti* are also present in *B. duncani* during its intraerythrocytic development as well as after cell division as links between daughter parasites (Figure S3). We do not maintain *Babesia* parasites other than *B. microti* and *B. duncani* and therefore we can't possibly determine whether the filamentous forms are found in every *Babesia* species in nature. However, a search of available Giesma-stained blood smears from other *Baebisia* species in the published literature revealed the presence of similar structures. We refer to these publications in the revised manuscript.

The BmGPI12 staining with confocal needs another control antibody to convince this reviewer.

Response:

We now provide supplemental images of BmGPI12-labeling by confocal microscopy using three peptide polyclonal (3 separate antibodies) and one monoclonal antibodies against BmGPI12. Data using these different antibodies are provided as supplemental figures (Fig. S2). The data with these antibodies reveal the same features reported in Fig. 1 with the polyclonal antibodies against the polypeptide (1-302).

Other points:

Figure 1, Figure S1A and Figure 5A look like different exposures of the same blot. The authors should supply a similar exposed version of the different blots to diffuse possible concerns.

Response:

We thank the reviewer for noticing this error, which occurred as a result of selecting which figures to be in the main paper and which figures to move to the supplementary files. We have now updated the manuscript and corrected the issue in Figure 1. In addition, we show the data for both *B. microti* strains LabS1 and PRA99.

Figure 3B and Fig S3B are duplicates.

Response:

Panel B of Figure S3 has now been removed from the revised version of the manuscript.

Figure 6B needs legend, how does export work?

Response: A legend for Fig. 6 has now been included in the revised manuscript and an explanation of the model is provided.

Figure S1 misses 'S' in second panel

Response: This figure is now as part of Figure 1 and the S has been added.

Authors need to mention that 3D EM would be beneficial to look at whole cell, e.g. FIB SEM or serial sections to investigate whether the vesicles are truly disconnected from the parasite body.

Response:

Thank you very much for the suggestion. We included a statement in the discussion to that effect.

Page 4, there are more than 3 species causing disease in humans.

Response: We have updated the manuscript and provide a reference to a recent review stating the presence of more species that infect humans (Krause. 2019).

Page 12, the protein is NOT found in the plasma fraction according to Figure S2. Please clarify. The labelling of the blots is not ideal.

Response:

BmRON2 most likely undergoes proteolytic degradation upon release of daughter parasites following rupture of the infected cell and we see the cleaved products in the parasite membrane fractions. A new Western blot has been included to show the presence of the full length protein and degradation products in the P and S (plasma) fractions. Degradation of BmRON2 has also been reported by Ord and colleagues (reference included in the revised manuscript).

Page 12: the Ord paper does not mention what it is referred at.

Response:

Ord et al. 2016 identified and characterized BmRON2 in parasite extracts. We referenced this publication in our manuscript as their results (BmRon2 profile in the parasite fraction) are similar to ours.

Page 14: what do the authors mean by 'erythrocyte environment'?

Response: this is basically the blood in mouse but since we are analyzing the plasma fraction, we refer to this environment as plasma throughout the manuscript.

Page 17: short term culture is not described in the methods

Response: Since we do not include data from short-term culture of *B. microti*, this sentence has been removed.

Page 18: maybe a concluding sentence or two at end of discussion?

Response: A concluding sentence at the end of discussion has now been added.

Reviewer #2 (Comments to the Authors (Required)):

The manuscript is interesting to explore how the antigen of *Babesia microti* is being transported to RBC membrane. However, I have some concerns in the manuscript as follows.

1. Two strains of *B. microti* were used in the present study. Furthermore, several knockout mice were used for maintaining of strains. However, I could find the reasons why two strains and knockout mice were necessary. Were there some differences in the development of parasitemia between immunocompetent and knockout mice, and results between two strains, respectively? The authors should add the information on these questions.

Response: For regular maintenance of *B. microti* strains we use both rag2D and scid mice upon availability. Parasite development in these immunocompromised mice is the same with parasitemia increasing over time and remaining around ~50% for several weeks.

2. For the determination of parasitemia, the sentence must be mixed up with some other method (p.6, lines 10-14).

Response: Thank you very much for pointing this out to us. In the revised manuscript, we have corrected this issue.

3. There was no data for "TER-119 was not identified in the plasma (p. 12, line 15) in Fig, 1B."

Response: Thank you very much for pointing this out. We corrected the typographical error to Fig. 1A.

4. The authors described that BmIPA48 was found inside of vesicles (p.15, line 11). However, some gold particles were found mainly outside of IV indicated by arrow in the upper left of Fig.5E.

Response: Thank you very much for pointing this out. The majority of gold particles appeared within vesicles (of the 8 vesicles in panel D, four have more than 4 gold particle, three have 3 gold particles and one has 2 gold particles). Gold particles that are outside are due to the fact that some of the vesicles are just below the focal plane in the section (outline of vesicles can be seen in the erythrocyte cytoplasm). The antibody most likely recognized the specific epitope although the vesicle as such is not visible in the section.

5. Maltese cross (Tetrad form) is a morphologic characteristic of *B. microti* and it appears in the early stage (less than 50% of whole parasitized RBC) before peak parasites and disappears in the later stage of developmental course of parasitemia in BALB-c mice (Infection and Immunity, 71:411-417, 2003). Although tetrad forms and tetrad forms with IOV were seldom found in the present study (Fig. 2C), the model of antigen export was suggested for tetrad form only in Fig. 6 without any explanations. Is this model applicable to paired from (two parasites)? If so, what were data for this model? The authors

should suggest a model based on the evidence.

Response: We have updated the model to include both the predominant form detected during parasite intraerythrocytic development (Ring and ring-like forms) as well as the tetrad stage, which is the late developmental stage once replication is completed. This stage, however, is short-lived and not often seen in blood smears from animals. We have also updated the description of the model in the manuscript.

Reviewer #3 (Comments to the Authors (Required)):

Thekkianiath et al present remarkable data suggesting novel mechanism of protein transport used by the intracellular parasite, *Babesia microti*. This parasite resides within the RBC but is not contained inside a vacuole. Its genome lacks homologs of the translocon machinery used for protein export by *Plasmodium falciparum*, a related Apicomplexan that also parasitizes RBCs but one that resides within a vacuole. The authors demonstrate, primarily through EM, that intracellular *B. microti* stages form a network of long tubes of incipient vesicles, termed ToV, that extrude from the parasite plasma membrane. Infected RBCs also contain vesicles containing parasite proteins. At least, two secreted proteins of *B. microti*, BP112 and IPA48, that elicit strong host antibody responses, are present in these ToVs and vesicles. Their immunolocalization is supported by cell-fractionation experiments. Localization of BP112 and IPA48 to ToV suggests that these structures are part of the parasite-encoded machinery for trafficking proteins to the host cell cytoplasm for eventual secretion. The EM data is compelling and supportive of the paper's conclusions.

One minor weakness that can be addressed in the Discussion section, is the relevance of the observation that IPA48 is not detected in the vesicle-free fraction. In this reviewer's opinion, this could result from lower sensitivity of the anti-IPA48 antibody compared to the anti-GP112 antibody or the higher overall abundance of GP112. Additional comments below are meant as suggestions to the authors.

1) It would be useful to test if ToV and vesicles were selective in their cargo. For example, testing the presence in ToVs or vesicle of parasite cytoplasmic proteins. If ToVs and vesicles serve in polar transport of secreted proteins, then parasite cytoplasmic proteins would be expected to be excluded from these structures. Furthermore, such "selection of proteins" could imply that *B. microti* uses a novel signal to mark proteins destined for secretion. On the other hand, if ToVs and vesicles bud off from the parasite plasma membrane in a more indiscriminate fashion, then they may contain both proteins destined for secretion and ones meant to be cytoplasmic.

Response:

Thank you very much for the suggestion. Testing if the TOVs and the vesicles are selective in their cargo is interesting to investigate in future studies. At this time, no antibodies against specific cargo molecules are available and such studies cannot be conducted at this time.

2) Is there any evidence for ToV merging with the RBC PM? It would be intriguing if the ToV directly deliver proteins to the RBC surface. What is the relationship between Bgp112-containing ToVs in Fig 4B and 4 E-F. Are the vesicles and tubes in Fig 4E-F thought to arise from the ToVs seen in Fig 4B?

Response:

Thank you very much for the question. Based on our results, we have not seen any evidence for TOV merging with the plasma membrane of red blood cells. Based on the method used to isolate vesicles, we believe that those vesicles seen in 4E and 4F represent the same IV and TOVs seen in 4B.

3) Is ToV formation associated with a specific parasite stage? While this seems to be the motivation behind Fig. 2C, the conclusion is not clearly stated.

Response:

Thank you very much for the question. Based on the Giemsa-stained smear results, we do not see any relationship between development stages and TOV formation. In the revised manuscript, we included this information.

4) Can the authors comment on how whether other Babesia species form TOVs?

Response:

Thank you very much for the question. So far, we only have access to *Babesia duncani*, which we can maintain in culture in human red blood cells (Abraham et al., 2018). Giemsa staining of *B. duncani*-infected red blood cells show the formation of filamentous forms during the parasite intraerythrocytic development. In the revised manuscript, we provide evidence for those TOVs seen in Giemsa-stained smears of *B. duncani* WA1 strain. Please see Fig. S3. Furthermore, we added references to published reports with Giemsa-stained images of other Babesia species that suggest the presence of these filaments in other Babesia species.

June 4, 2019

RE: Life Science Alliance Manuscript #LSA-2019-00382-TR

Prof. Choukri Ben Mamoun
Yale School of Medicine
Section of Infectious Diseases
15 York Street
Winchester Building WWW403D
New Haven, CT 6520

Dear Dr. Ben Mamoun,

Thank you for submitting your revised manuscript entitled "Evidence for vesicle-mediated antigen export by the human pathogen *Babesia microti*". Your work was re-assessed by original reviewer #1 and given this reviewer's input, we would be happy to publish your paper in Life Science Alliance pending final revisions:

- please address the remaining concerns of reviewer #1 by further discussion/changing the discussion and adding further context from the existing literature
- please upload the supplementary figures as individual files; these will be displayed in-line in the HTML version of your paper, so please provide them as single page files (figure S2 currently spans over two pages)
- please link your profile in our submission system to your ORCID iD, you should have received an email with instructions on how to do so

A. FINAL FILES:

B. MANUSCRIPT ORGANIZATION AND FORMATTING:

Sincerely,

Reviewer #1 (Comments to the Authors (Required)):

The authors addressed most comments, however introduced one major flaw. It appears the authors did not understand one of my questions and unfortunately also answered this question incorrectly, which is concerning. The statement 'Examination of previously published images of Giemsa-stained blood smears with *Babesia microti* or other *Babesia* species (*B. divergens* or *B. bovis*) revealed the presence of similar structures during these parasites' intraerythrocytic development ((Hildebrandt et al., 2013; Sevilla et al., 2018; Vannier and Krause, 2018))' is not correct. The images in the respective papers clearly show that *B. divergens* and *B. bovis* do NOT show any of the structures reported for *B. microti*. This absolutely needs to be stated VERY clearly. The authors should cite work e.g. Kühni-Boghenbor et al Cell Micro 2012 on *Theileria*, where similar (yet different) structures have been observed. As *Babesia* and *Theileria* are close relatives a brief discussion/speculation should be added how/why some species make these extensions and others not. A discussion on the divergence within the Plasmodium clade could help.

June 6, 2019

RE: Life Science Alliance Manuscript #LSA-2019-00382-TRR

Prof. Choukri Ben Mamoun
Yale School of Medicine
Section of Infectious Diseases
TAC Building
TAC-S215
New Haven, CT 6520

Dear Dr. Ben Mamoun,

Thank you for submitting your Research Article entitled "Evidence for vesicle-mediated antigen export by the human pathogen *Babesia microti*". I appreciate the introduced changes and it is a pleasure to let you know that your manuscript is now accepted for publication in Life Science Alliance. Congratulations on this interesting work.

DISTRIBUTION OF MATERIALS:

Again, congratulations on a very nice paper. I hope you found the review process to be constructive and are pleased with how the manuscript was handled editorially. We look forward to future exciting submissions from your lab.

Sincerely,

Andrea Leibfried, PhD
Executive Editor
Life Science Alliance
Meyerohofstr. 1
69117 Heidelberg, Germany
t +49 6221 8891 502
e a.leibfried@life-science-alliance.org
www.life-science-alliance.org